# Concept of Orodispersible or Mucoadhesive “Tandem Films” and Their Pharmaceutical Realization

**DOI:** 10.3390/pharmaceutics14020264

**Published:** 2022-01-22

**Authors:** Anja Göbel, Jörg Breitkreutz

**Affiliations:** 1Institute of Pharmaceutics and Biopharmaceutics, Heinrich Heine University, Universitätsstr. 1, 40225 Düsseldorf, Germany; anja.goebel@hhu.de; 2Sapiotec GmbH, Nikolausstr. 18, 97082 Würzburg, Germany

**Keywords:** personalized medicine, oral film preparations, incompatibilities, mucoadhesive buccal films, solvent casting

## Abstract

Orodispersible or mucoadhesive films as a patient-oriented dosage form for low-dosed drugs are usually produced using solvent casting. This paper presents a modification of the solvent casting technique that aimed to divide oral films into two or more compartments. The proposed objectives and fields of applications include improved handling properties and safety of application, the optimization of drug release kinetics and the enhancement of long-term stability when combining two or more active pharmaceutical ingredients into one oral film. A feasibility study for the combination of different film-forming polymers to generate the so-called tandem films was performed. As examples of practical implementation, orodispersible applicator films consisting of a drug-loaded section and a handheld piece were cast, and mucoadhesive buccal tandem films were cast to optimize the dissolution rate of the films.

## 1. Introduction

In the past few years, drug-loaded orodispersible or buccal films have gained increased interest as an alternative dosage form that is conveniently administrated compared to conventional solid drug forms, such as tablets or capsules. Orodispersible films (ODFs) offer major advantages, especially for patients with difficulties swallowing, such as the pediatric or geriatric population. When administered, the dosage form dissolves immediately in the oral cavity and can be swallowed together with the saliva. Buccal films can reach a fast onset of action and circumvent the first-pass effect if the drug substance permeates the oral mucosa and is taken up into the bloodstream. Commonly, oral films are produced using the solvent casting technique. This technique involves the fabrication of a polymer dispersion or solution with suspended or dissolved active pharmaceutical ingredient(s) (API(s)). Excipients, such as plasticizers, colorants and flavors, may be added as well [1], and the polymeric formulation is then cast on a coating bench. The cast film layer is dried and then cut into the individual films, with the most common sizes being 2 × 2 or 2 × 3 cm [2]. In the literature, variations of this process and concepts for the production of multi-component films were described: films consisting of two or more layers were introduced in order to improve the stability of the API(s) [3]. In a recently approved film formulation, the tolerability and shelf life of the drug could be optimized by casting layers with different pH values [4,5]. Furthermore, a backing layer preventing the release of an API into the saliva was realized [6]. One drawback of the production of multilayer films is an increased expenditure of time. In addition, when casting a second film layer on top, the lower layer can (partially) dissolve, and thus lead to a mixing of the layers. This can make the separate casting of two layers necessary, which are then laminated onto each other [6]. In this work, an alternative concept for the fabrication of multi-component films is presented. By modifying the coating blade via implementing subdivisions, it was possible to cast two or more polymer solutions or suspensions in parallel. This may result in a striped polymer layer that can subsequently be cut into bi- or multi-compartmental films after drying, which is referred to as “tandem films” in the following. Different concepts for an application of such tandem films are presented. Two of these possibilities (referred to as concept (a) and concept (b) in the following) were further investigated and their practical implementation is described in this publication.

The proposed possibilities and advantages of such tandem films include are shown in Figure 1.

### 1.1. Concept (a): Preparation of Applicator Films

The aim of this approach was to couple a drug-loaded ODF to a drug-free “applicator”. One possibility is an applicator that is insoluble in human saliva. The hand-held applicator can either be intended to be pulled off from the drug part with the lips or the film is applied by placing the drug-containing part onto the tongue, where it dissolves immediately and the applicator can be disposed of afterward. This could be an easy and more convenient way of administration for young children and patients with limited dexterity since the small and thin ODFs might be difficult to handle for these patients. Alternatively, the drug-free applicator can be soluble and rapidly disintegrates as well, allowing the intake of the complete film. This approach could help to make the application of films containing highly potent ingredients more safe since the administering person (patients, parents or nursing staff) can avoid touching the drug-loaded film part. Solid dosage forms containing potentially hazardous ingredients, such as hormonal or antineoplastic drugs, are usually covered with a film coating (tablets) or a drug-free shell (capsules), which prevents residuals of the API from being transmitted onto the hands or gloves of the administering person. Separating the film into a drug part and a drug-free applicator, that can be distinguished by color might reduce the carryover of API(s) into the environment and onto other persons.

### 1.2. Concept (b): Optimization of the Dissolution Properties

Buccal film formulations enable the permeation of an API through the oral mucosa, which leads to a rapid onset of action and offers the possibility to avoid the first-pass effect by circumventing the enterohepatic circulation [7]. In previous studies, polyacrylic acid (PAA) was shown to be a potent enhancer of the mucoadhesion of hypromellose-based film or tablet formulations [8,9]. It was shown to be superior in increasing the bioadhesive force compared with other commonly used ionic mucoadhesives, such as chitosan and sodium alginate [9]. However, PAA is known to slow down the dissolution kinetics of mucoadhesive buccal films [10], which can be a problem if a fast onset of action is desired. The addition of electrolytes decreases the sustaining effect, but with that, also the mucoadhesive strength [11]. To make use of the mucoadhesive properties of PAA while maintaining a rapid drug dissolution, the tandem film technique was used within the scope of this study by dividing the film into drug-loaded (rapidly dissolving) and mucoadhesive segments.

Another possible application is the combination of different release kinetics in one film: the implementation of different release kinetics in one dosage form containing one or more API(s) was successfully realized in various solid dosage forms. Examples are tablets with two active ingredients consisting of an immediate release and a sustained release layer [12,13,14], matrix tablets with an initial boost or multilayered pellets that can, for example, be filled into capsules [15]. Literature that describes the realization of different release kinetics from one film formulation is scarce. Mucoadhesive film or patch formulations with a prolonged release are described in the literature [16,17], although their fabrication can be challenging due to their high surface area and low thickness. An alternative attempt to control the drug release properties from oral films was realized by the implementation of sustained-release micropellets into rapidly disintegrating films [18]. The tandem film technology offers the potential to combine different release kinetics in one dosage form by casting multicompartmental films with an appropriate composition of polymers in the individual areas.

### 1.3. Concept (c): Tandem Films for the Combination of Incompatible APIs

For a broad range of diseases, such as arterial hypertension, Parkinson’s disease or HIV, a combination of different APIs can be favorable in order to improve the efficacy of therapy and reduce the side effects [19,20,21]. However, complicated dose regimens with a high number of medications and daily intakes can reduce the patients’ adherence [22,23,24] and, therefore, may hamper the therapy’s success. Therefore, it is beneficial to apply drugs that combine different APIs in one dosage form. However, developing such combination drugs may be challenging due to possible incompatibilities. These can occur, for example, due to the ionic interactions of the APIs and incompatibilities with excipients and/or stability optima in different conditions. When water is used as a solvent to cast oral films, remaining water contents between 2 and 11% are commonly reported depending on the film formulations and the method applied [25,26,27]. Hence, when developing orodispersible or buccal films, the resulting pH value in the polymer matrix can play an important role. It can be necessary to separate APIs from each other to improve their long-term stability. In the past, attempts were made to realize such a separation by casting bi- or multilayer films [3]. However, the high contact area and low layer thickness were found to be critical with regard to the migration of API(s). Moreover, problems such as partial dissolution of the bottom layer when casting the second one, insufficient adhesion between the layers and shrinking of the layers were reported [3]. Tandem films might be advantageous due to the smaller contact area between the compartments and reduce the risk of migration of one API into the other part. Furthermore, their production has the potential to save time because an intermediate drying step of the bottom layer is not necessary.

## 2. Materials and Methods

### 2.1. Materials for Film Casting

Hypromellose (HPMC, Pharmacoat^®^ 603/606, Shin Etsu, Chiyoda, Tokyo, Japan; substitution type 2910), hyprolose (HPC, Nisso HPC SL, Nippon Soda, Tokyo, Japan; molar substitution 3.62 [28]), polyvinyl alcohol (PVA, PE-05JPS, JVP, Osaka, Japan; degree of hydrolysis 88%), pullulan (Tokyo Chemical Industry, Tokyo, Japan), ethyl cellulose (EC, AqualonTM N10/N22, Ashland, Wilmington, DE, USA; ethoxyl substitution 48.0–49.5%) and polyacrylic acid (Carbopol 971P NF, Lubrizol, OH, USA) were used as polymeric film formers. Anhydrous glycerol (Caelo, Hilden, Germany) and triethyl citrate (TEC, Citrofol^®^ AI, Jungbunzlauer, Swizerland) were used as plasticizers. Tragacanth (Merck Millipore, Burlington, MA, USA) was used as a thickening agent and microcrystalline cellulose (MCC, Vivapur 101, JRS Pharma, Rosenberg, Germany), sodium starch glycolate (SSG, Vivastar P, JRS Pharma), povidone (PVP, Kollidon 90 F, BASF, Ludwigshafen, Germany), crospovidone (PVP-Cl, Kollidon Cl, BASF) and sodium croscarmellose (CMC, Ac-Di-Sol SD-711, DuPont de Nemours, Wilmington, DE, USA) as disintegrants. Deionized water and ethanol of analytical grade were used as solvents. Bisoprolol hemifumarate (Arevipharma GmbH, Radebeul, Germany) and theophylline monohydrate (BASF) were used as active ingredients. Amaranth (BASF) was used as a colorant and disodium hydrogen phospahe (Carl Roth, Karlsruhe, Germany) and phosphoric acid 85% (Sigma-Aldrich, Steinheim, Germany) as buffering agents.

### 2.2. Solvent Casting

To generate the pre-casting masses, the components were mixed until reaching homogeneity on a magnet stirrer (RCT classic, IKA, Breisgau, Germany) or overhead stirrer (Eurostar 20 digital, IKA, Staufen, Germany). All films were cast on an automatic film applicator (Coatmaster 510) with a coating knife with an adjustable blade height (Multicator 411, both by Erichsen, Hemer, Germany) and a polyamide-coated polyester liner (Mediflex^®^ XM AMWL (45/105), Amcor Flexibles, Ghent, Belgium). The standard casting speed was 6 mm/s and 40 °C was used as a drying temperature, unless stated otherwise.

### 2.3. Construction and Evaluation of Prototype Models for Subdivision of the Coating Blade

Two different models of subdivided coating knives were made of a stainless steel strip of 20 mm width and 0.5 mm in thickness via spot welding. Both stainless steel constructions divided each coating knife into subdivisions of 30 mm and differed only in regard to their leading edge: the subdivisons from model 1 sat flush with the rectangular frame of the insert, while an overhang of 8 mm in length was present in model 2 (Figure 2a,b). The aim was to prevent the mixing of both pre-cast solutions in the gap in front of the coating blade. A third insert was printed from polylactic acid (PLA) or acrylnitril butadiene styrol (ABS) filaments with a 3D printer (Prusa i3 MK3, Prusa Research, Prague, Czech Republic). This model consisted of two holders with notches at 5 mm intervals, which could be attached to the coating knife, and the actual subdivisions, which could be inserted at variable intervals (Figure 2c). Some of the PLA subdivisions were additionally coated with a thin layer of epoxy resin (XTC-3D™, Smooth-On, Macungie, PA, USA) to smoothen the surface. The quality of separation of all models was tested using a parallel cast of different HPMC solutions, as described in Table 1. The HPMC solutions consisted of 15% HPMC, 3% glycerol and water. Two of the solutions were dyed with 0.15% of a 3% aqueous amaranth solution (*w*/*w*).

The evaluation of the durability of the materials from which the subdivided coating blades were made was based on literature data, as well as previous experience. For the assessment regarding resistance to thermal stress, acidic/alkaline pH and mechanical stress, only those parameters were taken into account that were seen as relevant for a solvent casting process. A temperature range of 15–80 °C; a physiologically tolerable pH; ranging from 3–9; and isopropanol, ethanol and acetone as commonly used solvents for a solvent casting process were considered.

### 2.4. Preparation of Tandem Films Made from Different Polymers

To manufacture tandem films consisting of solutions of different polymers (Table 2), the PLA subdivision setup was used.

All films consisted of 20% of one of the following film-forming agents: HPMC, HPC, PVA, pullulan or EC. The type and amount of plasticizer were adapted based on preliminary studies to obtain flexible and non-sticky films. Pullulan as a film former led to very low viscous solutions in preliminary studies and is often combined with thickening agents [29,30]. A very low viscosity (<1 Pa·s) was found to increase the risk of coalescence and mixing of the two polymer solutions during the casting and drying of tandem films, although this was dependent on the individual polymers. Tragacanth in a concentration of 0.2% was chosen as a thickener for the pullulan formulation. Deionized water was used as a solvent or, in the case of the EC formulation, a mixture of ethanol and water. All pre-casting solutions were cast both as conventional and tandem films with a 500 µm gap width.

### 2.5. Solubility Parameter Calculations

The partial solubility parameters (dispersion parameter (*δd*), polarity parameter (*δp*) and the parameter associated with hydrogen bonds (*δh*)) of the film-forming polymers were calculated based on the group contribution parameter set developed by Just et al. [31]. The calculation of these parameters is based on the functional groups present in the molecule, their proximity to other groups and the molecule volumes. Therefore, structurally similar molecules should have similar solubility parameters. The volume-dependent solubility parameter (*δv*, Equation (1), [32]) and the total Hansen solubility parameter (*δt*, Equation (2)) of the film-forming polymers were calculated based on the following equations:(1)$$\delta v=\sqrt{\delta d^{2}+\delta p^{2}},$$
(2)$$\delta t=\sqrt{\delta d^{2}+\delta p^{2}+\delta h^{2}}.$$

Hence, the volume-dependent solubility parameter *δv* is a simplification that includes both the disperse and polar group contributions. It was used for visualization of the solubility properties of molecules in a two-dimensional graph, which is called the Bagley diagram [32]. For further details, such as the group contribution parameter set, we refer to [31]. The molar weights for the calculations were as follows: 35,000–135,000 for HPMC, 100,000 for HPC, 200,000–600,000 for pullulan and 140,000 for EC [33,34,35,36,37]. A total of 500–5000 repetitive units were assumed for PVA [38], giving a molecular weight of 24,600–245,300.

### 2.6. Dynamic Viscosity

The dynamic viscosities of the pre-casting solutions were measured with a rotational rheometer (Kinexus pro, Malvern Panalytical, Worcestershire, UK) equipped with a cone (Ø 60 mm, 1°) and a plate (Ø 60 mm). A linear shear ramp from 0/s to 20/s was run for 5 min at 40 °C and the dynamic viscosity was determined at a shear rate of 12/s (*n* = 6), as it complied with the shearing conditions on the coating applicator at a 6 mm/s casting speed and for a 500 µm gap width.

### 2.7. Basic Film Characterization and Wettability

The dry film thickness and film mass were determined based on 2 × 3 cm pieces. The folding endurance was determined by folding and unfolding the film ten times or until it broke. The wettability of the films was determined with a drop shape analyzer (DSA100, Krüss, Hamburg, Germany) by dosing 12 µL of deionized water onto the surface of the film samples (*n* = 6). Images of the droplets were taken using the integrated camera and the contact angle was determined after 5 s. No swelling of the films was observed within this short time.

### 2.8. Determination of the Films’ Mechanical Strength

To examine the tensile properties of the films, they were cut to a size of 7.5 × 1.25 cm. This corresponds to specimen type 2 of the DIN EN ISO 527-3 for the determination of the tensile properties of films and sheets, reduced by 50% due to the limits of the small-scale tensile tester [39]. The tensile strength and elongation to break were determined using a texture analyzer (TA.XT2i, Stable Micro Systems, Surrey, UK) with miniature tensile grips (*n* = 6). An initial distance of 49 mm and a speed of 0.1 mm/s were set and the measurement was stopped when the film was ripped apart.

### 2.9. Preparation of Applicator Films

For the preparation of applicator films with a soluble or insoluble applicator, bisoprolol was chosen as an API commonly used in the treatment of arterial hypertension. This particularly affects elderly patients [19]. Bisoprolol is administered in a daily dose of 5–20 mg [40], making it particularly suitable for orodispersible films. Five disintegrants were tested during the formulation development in order to achieve rapid disintegration. The pre-casting dispersions were prepared as shown in Table 3, cast at a 650 µm blade height and evaluated for their disintegration properties with the “slide frame and ball” method [41]. The EC applicator solution was prepared with 30% EC (20% Aqualon^TM^ N10 + 10% Aqualon^TM^ N22), 10% triethyl citrate, 5% deionized water (*w*/*w*) and ethanol. A pre-casting solution for a dissolvable applicator was prepared analogously to formulation 7 (Table 3) by replacing the API with water and adding 0.15% of a 3% aqueous amaranth solution (*w*/*v*). Tandem films were cast by combining drug solution 7 with one of the applicator solutions. They were cut into a size of 2 × 3 cm (films with orodispersible applicator) or 2 × 5 cm (films with insoluble applicator).

### 2.10. Bisoprolol Content Determination

The bisoprolol content of the applicator films was determined via an HPLC apparatus (Agilent 1260 Infinity, Agilent Technologies, Santa Clara, CA, USA) equipped with a C18 with column (125 mm × 4.6 mm, 5 µm particle size, Eurospher II 100-5 C18 A, Knauer, Berlin, Germany). The method used was derived from the HPLC method described in the European pharmacopeia monograph [42], with adaptions only in the column and gradient (Table 4) to shorten the run time. The adapted method showed good linearity (R^2^ > 0.9995) and injection precision (RSD < 1%).

### 2.11. Preparation of Mucoadhesive Tandem Films for Rapid Release and Dissolution Testing

To examine the possible benefits of mucoadhesive buccal films (MBFs) prepared as tandem films, different film preparations containing theophylline as a model drug and HPMC or HPMC + polyacrylic acid (PAA) as film-forming polymers were prepared as depicted in Table 5. The formulations were cast into one-compartmental films (A(v), B(v), C(v)) and tandem films (A(v) + B(p) and A(v) + C(p)). The casting speed was adapted to 2 mm/s for this part of the study. All films were cut to a size of 2 × 3 cm.

### 2.12. Dissolution Testing and Content Determination and of Mucoadhesive Films

The dissolution properties of the MBFs were evaluated with the slide frame method, as described by Krampe et al. [43]. This method simulated the biorelevant conditions by taking the human mouth temperature, the saliva’s pH and buffering capacity and the average saliva flow into account [44,45]. Every 30 s, 250 µL phosphate buffer (pH 7.35, 67 mosmol/kg, 37 ± 0.5 °C) were taken from the vessel filled with 400 g buffer solution and pipetted on a 2 × 3 cm film sample that was placed on a paper filter (Figure 3). A UV probe with a light source and spectrometer (T300-RT-UV-VIS + DH-2000-BAL + USB 4000, Ocean Insight, Orlando, FL, USA, detection wavelength 270 nm) was used to measure the UV absorption of theophylline (*n* = 3). In addition, the theophylline content was determined with the same setup after dissolving the films in 500 g of buffer solution (*n* = 3–5).

## 3. Results and Discussion

### 3.1. Assessment of Model Constructions for the Preparation of Tandem Films

Two-compartmental tandem films could be successfully cast with both stainless steel constructions as well as with the 3D-printed models (Figure 4). For the HPMC solutions containing Pharmacoat^®^ 606, viscosities of 0.96 ± 0.06 Pa·s (HV) and 0.97 ± 0.10 Pa·s (HV red) were determined, while the solution with Pharmacoat^®^ 603 (LV red) had a lower viscosity of 0.19 ± 0.02 Pa·s. For the combination of HPMC polymer solutions of the same type and nominal viscosities, no differences in the separation precision were visible, depending on the utilized subdivision model. For the combination of HPMC solutions with different viscosities, no sufficient separation was achieved with the precursor stainless steel model due to the mixing of both polymer solutions in the gap in front of the coating blade. Using the optimized stainless steel insert and the 3D-printed inserts led to an improved separation, but some separation lines still appeared blurred. Similar and sufficiently high viscosities (about 1 Pa·s) of the polymer solutions were concluded to improve the separation.

No differences in the quality of separation were observed, depending on the material of the 3D-printed inserts. Both ABS and PLA are soluble in strong acids or bases but stable in diluted acids/bases (Table 6, [46]), making the use of pre-cast solutions with physiologically tolerable pH values non-critical. PLA shows a higher resistance toward organic solvents such as acetone [46], which might be relevant in the case of pre-casting formulations based on organic solvents or cleaning purposes. ABS is less brittle than PLA [47] and is more temperature resistant compared with PLA (glass transition temperature 100–110 °C and 60–65 °C, respectively [48]), which may be relevant when higher drying temperatures are applied. Typically, drying temperatures between 30 °C and 80 °C are described for the solvent casting of oral films [49,50,51], or films are dried at room temperature [52]. In this study, the drying temperature was limited due to the cloud point of cellulose ethers, such as HPMC and HPC, which leads to precipitation at higher temperatures [53,54]. In (semi)continuous coating machines, the coating blade is usually static and the liner is conveyed through a drying oven [55], which makes the heat resistance of the coating knife less relevant. The custom manufacturing of the subdivision made of PLA or ABS via 3D printing was simple, cost-efficient and quick; furthermore, it enabled the adaption of the segment width, but the materials were less resistant compared with the stainless steel models. It should be considered that a change in coating thickness makes an adaption of the inserts necessary due to the changed gap size.

Based on the presented results regarding the quality of separation and material properties, the type of insert model for the production of tandem films should depend on the application purpose. For preliminary studies and formulation development, the 3D-printed inserts were considered the superior choice due to their fast and easy fabrication and, therefore, higher flexibility in segment width and coating thickness. Stainless steel offers better durability regarding resistance to heat and organic solvents and is therefore considered to be the most suitable material for routine productions.

Since the possibility of an easy adaption of blade height was crucial for the experimental work and high resistance with regard to ethanol and acetone was desired, the 3D-printed PLA model coated with epoxy resin was chosen for the subsequent experiments.

### 3.2. Preparation and Subsequent Evaluation of Tandem Films Based on Different Film Formers

The dynamic viscosities of the five different pre-casting solutions made of HPMC, HPC, PVA, pullulan and EC ranged from 0.8 Pa·s to 4.2 Pa·s (Table 7). This was seen as sufficiently high for the preparation of tandem films. The films cast from the five different film-forming polymers were smooth, flexible (folding endurance ≥ 10) and transparent.

The tensile strengths of the HPMC, HPC, pullulan and EC formulations were between 5 and 30 N/mm^2^, and the elongation to break was between 15 and 30% (Figure 5). The PVA films did not rip until the tensile tester’s maximum distance was reached (elongation > 300%). A direct comparison of the film-forming polymers’ properties is limited due to the fact that different types and quantities of plasticizer were necessary to obtain flexible and non-sticky films. This can have a significant effect on the elongation to break and the tensile strength of film formulations [57,58,59]. Five out of 10 polymer combinations resulted in successful tandem films. It can be clearly seen that the tensile strengths of all tandem film formulations were reduced compared to the original film formulations. Apparently, the intermolecular interactions between the different polymers were less strong than the cohesive forces between the molecules of the same polymer. This result matches the observations that all tested tandem film strips ripped at the interface of the two phases and that there was a sharp separation of the polymer solutions within the tandem films (see Figure 6). Hence, the pre-casting solutions´ measured viscosities of 0.8–4.2 Pa·s were considered appropriate to flow together into tandem films without mixing too much. The pre-casting solution based on pullulan was the only formulation that could not be combined with other polymers to form coherent tandem films, as the dried and cut films already tore when detaching from the liner was attempted. It has to be remarked that tragacanth as a thickening agent had an impact on the film´s properties as well and might negatively affect its mechanical stability in the case of incomplete miscibility. Pullulan formulations with no thickening agent had a low viscosity, which led to the coalescence of the solutions during tandem film casting in pretests.

Out of the successful polymer combinations, the combinations of HPMC + HPC and HPMC + PVA resulted in the films with the highest tensile strengths.

Even though HPMC and HPC are structurally similar, the polymers were found to be immiscible when evaluated using differential scanning calorimetry [60]. Miscibility studies of HPMC and PVA based on viscosity, density, refractive index and ultrasonic velocity indicated miscibility if the HPMC ratio was 60% or higher [61]. Prassad et al. found partial miscibility of pullulan and HPMC blends depending on the ratio of both excipients but immiscibility if the HPMC ratio was below 50% [62]. Parts of these results are in contradiction with the results in this study, as tandem films from HPMC and HPC showed the highest tensile strength, which indicated mixing of the pre-casting solutions and, presumably, the formation of hydrogen bonds. The comparison of three-dimensional solubility parameters calculated using the group contribution parameter set by Just et al. [31] shows that HPMC, HPC, PVA and EC are located close to each other in the Bagley diagram with similar *δv* values (Figure 7), while pullulan is located a large distance away. Typically, similar solubility parameters—and thus a smaller distance in the Bagley diagram—indicate better miscibility. This matches the observation that pullulan was the only polymer that could not be combined into tandem films with the other four film formers. However, it does not explain why HPC and EC could not be combined, as they are positioned close to each other in the diagram. Maybe the strong adhesion of the HPC film to the polyamide liner favored breaking during the detaching of the tandem film. For those polymers for which no exact molar mass was specified, inserting the upper and lower limits of the span into the calculations changed the parameters´ values by not more than 0.03 MPa^0.5^.

### 3.3. Concept (a): Preparation of Applicator Films

The disintegration times of the evaluated bisoprolol-loaded pre-formulations and the optimized formulation can be seen in Figure 8. All disintegrants, except for PVP Cl, reduced the mean disintegration time and the SSG film exhibited the fastest disintegration. However, the mean disintegration time of 73 s was still considered too high. Although there is no uniform method or specification for the determination of the disintegration time of ODFs, a limit of 30 s (as it is recommended for orodispersible tablets by the FDA [63]) is often applied [64]. Changes in the formulation, such as the reduction of the HPMC quantity or its molecular weight, led to a reduced disintegration time but resulted in brittle films that were considered unsuitable for further use (not shown). In the optimized SSG film, air was selectively introduced through rapid stirring (overhead stirrer with spiral attachment, 1000 rpm, 2 min), which resulted in a porous film with an improved disintegration time. Due to the higher volume and lower density of the pre-casting solution, the bisoprolol content in the formulation had to be adapted in order to obtain the target dose of 5 mg/4 cm^2^. The optimized SSG formulation exhibited a sufficient folding endurance of >10, as well as a disintegration time of <30 s.

Applicator films both with an insoluble applicator for easier administration (Figure 9a) and with an orodispersible applicator for safe application of highly potent APIs (Figure 9b) were successfully cast. Each production was carried out twice to examine the reproducibility. The drug-containing sections and applicators were sharply separated from each other, visible in the magnified images. The films with an EC applicator and a drug-containing part of 2 × 2 cm had a drug load of 4.9 ± 0.16 mg and 5.2 ± 0.18 mg bisoprolol fumarate per film (*n* = 10, mean ± sd) and, therefore, fulfilled the requirements of the European pharmacopeia (uniformity of dosage units) for a film containing 5 mg bisoprolol fumarate [65]. The film with a soluble applicator had a lower area of 1.5 × 2 cm drug-containing part and had a lower drug content of 2.9 ± 0.06 and 2.9 ± 0.11 mg (*n* = 10, mean ± sd). The utilization of amaranth as a colorant enabled an easy distinction between the drug-loaded part and the applicator part.

### 3.4. Concept (b): Preparation of Mucoadhesive Tandem Films with Rapid Dissolution Properties

The films/film parts containing theophylline appeared white due to the suspended API, while the mucoadhesive, PAA-containing parts of the tandem films were slightly opaque but particle-free (Figure 10). All film formulations led to flexible films with evenly distributed theophylline contents (RSD < 5%) and were chosen for subsequent dissolution tests.

Films containing PAA as an enhancer of mucoadhesion had a lower theophylline load per 6 cm^2^ film compared with the formulation based on HPMC, presumably due to their higher resulting viscosities. Increasing viscosities of the pre-casting masses are known to lead to a higher deviation between the gap height and the actual wet film thickness, which reduces the drug load [66]. For the tandem films, the resulting surface of the drug-loaded film was reduced to 3 cm^2^. It was observed that the “stripe” shape led to a smaller drug load compared with the “band-aid” shape, which cannot be explained only by the space occupied by the coating knife insert. Instead, it can be assumed that the sticking of the pre-casting mass to the insert and resulting capillary forces between the subdivisions led to a lower thickness and, consequently, to a lower drug load. These effects should be considered and can be overcome by adapting the blade height of the coating knife.

The addition of PAA as an enhancer for mucoadhesion led to a slower dissolution profile in the one-compartmental buccal films (Figure 11), which is disadvantageous when a rapid onset of action is desired. Less than 50% of the API was dissolved within 45 min. Decreasing the amount of PAA in the formulation (formulation B) did not improve the dissolution rate.

Separating the film into a central theophylline-loaded part with drug-free edges containing the mucoadhesive PAA (“band-aid” shape) or into a multicompartmental film (“striped” film) was feasible. The resulting tandem films showed faster dissolution compared with the one-compartmental film formulations. After 20 min, more than 90% of the theophylline was dissolved from formulations A + B/1 and A + B/2, and more than 70% from formulation A + C/1. To ensure the unidirectional release of the API through the mucosal membrane, it is possible to implement an additional backing layer [6]. This backing layer could, for instance, consist of a mixture of HPMC and PAA, as this combination was shown to be an effective barrier that slows down the drug release.

## 4. Conclusions

Concepts for the preparation of tandem films and their possible applications were developed and their feasibility was successfully evaluated and demonstrated. Different constructions for subdivisions of the coating blade made of PLA, ABS and stainless steel were designed, built and evaluated. Moreover, combinations of five different film-forming polymers that are commonly used for the preparation of orodispersible or mucoadhesive buccal films were examined. Out of three presented concepts for tandem films and their advantages over conventional one-compartmental films, two approaches were realized within the scope of this study: it was feasible to design applicator films consisting of a soluble or insoluble drug-free handheld piece and a rapidly disintegrating drug-loaded part. This could be an important approach to improve the handling properties of thin films and the safety of application when administered by others. Furthermore, mucoadhesive buccal tandem films were designed. Improved dissolution properties of these tandem films compared to one-compartmental mucoadhesive films were shown. For further transfer of the technique to production scale, reproducible and precise cutting of the tandem films is essential in order to ensure sufficient content uniformity. At the laboratory scale, sufficient content uniformity of the cut tandem films was successfully achieved. A limitation of the tandem film technique that remains is the limited maximum drug load due to the reduction in the available surface area, which restricts the application to highly potent APIs.

## Figures and Tables

**Figure 1 pharmaceutics-14-00264-f001:**
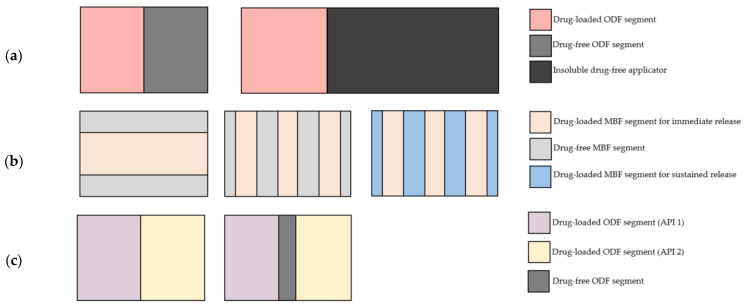
Conceptualization of different tandem films and their possible applications: (**a**) applicator films, (**b**) optimization of dissolution rate and (**c**) optimization of stability in the case of incompatible APIs.

**Figure 2 pharmaceutics-14-00264-f002:**
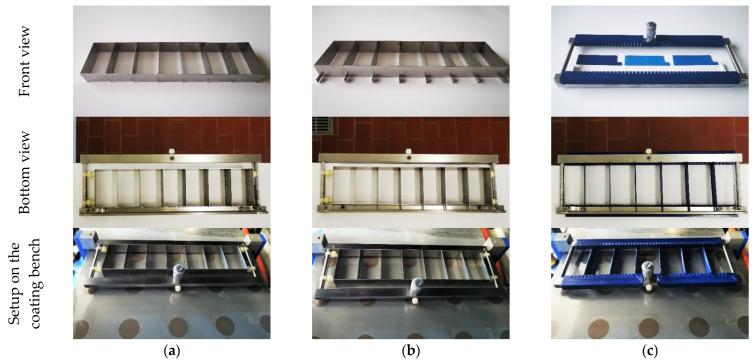
Comparison of the insert types in the front and bottom views and installed on the coating bench: (**a**) precursor steel model, (**b**) optimized steel model and (**c**) 3D-printed model.

**Figure 3 pharmaceutics-14-00264-f003:**
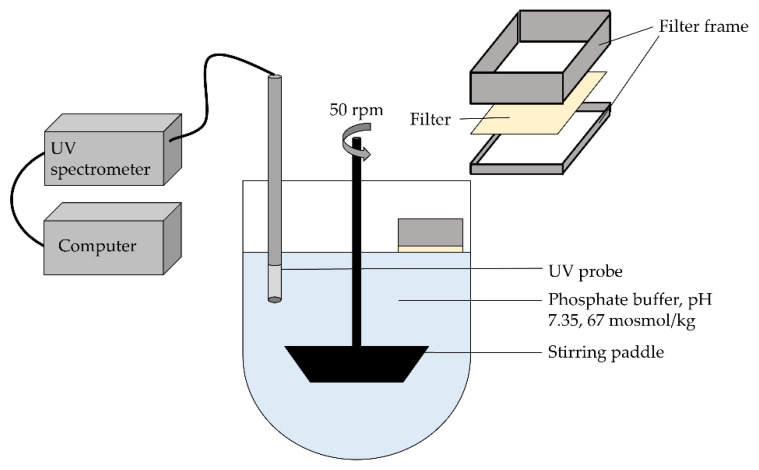
Setup for dissolution testing of the oral films, modified from Krampe et al. [43] with kind permission from Elsevier.

**Figure 4 pharmaceutics-14-00264-f004:**
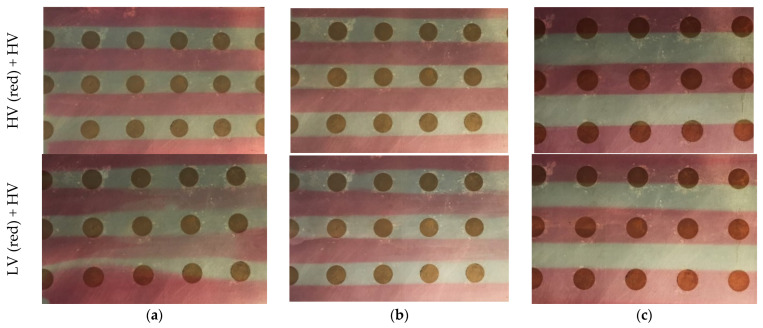
Photographies of the placebo tandem films cast of higher viscous (HV) and lower viscous (LV) HPMC solutions with the three insert models. The visible circles are for vacuum extraction. The films were cast with a (**a**) precursor steel model, (**b**) optimized steel model and (**c**) 3D-printed PLA model.

**Figure 5 pharmaceutics-14-00264-f005:**
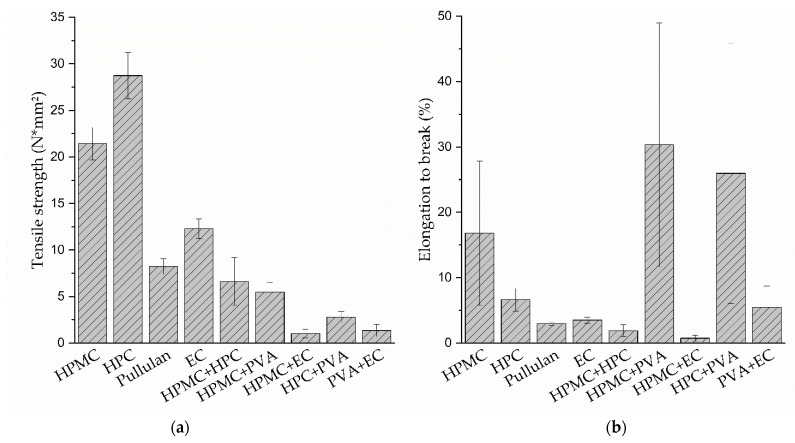
(**a**) Tensile strength and (**b**) elongation to break of the original formulations and the successfully prepared tandem films (*n* = 6, mean ± sd). The PVA films did not rip until reaching the tensile tester´s maximum distance (elongation > 300%).

**Figure 6 pharmaceutics-14-00264-f006:**
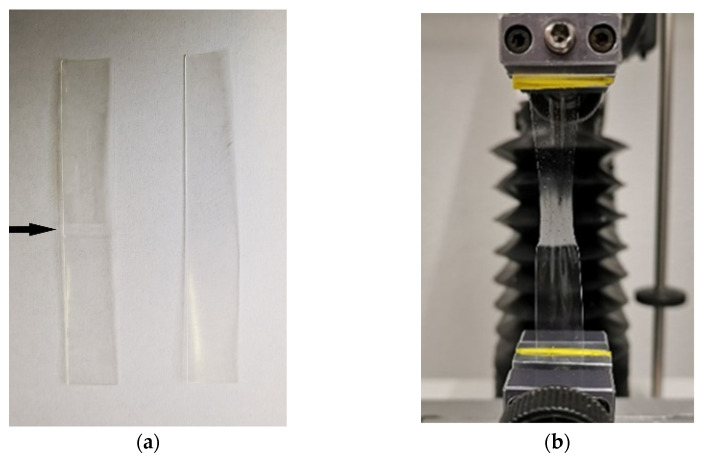
(**a**) Cut-to-size strips for tensile testing (left: HPMC + HPC, arrow marks the polymer junction, right: HPC) and (**b**) tandem film during the measurement.

**Figure 7 pharmaceutics-14-00264-f007:**
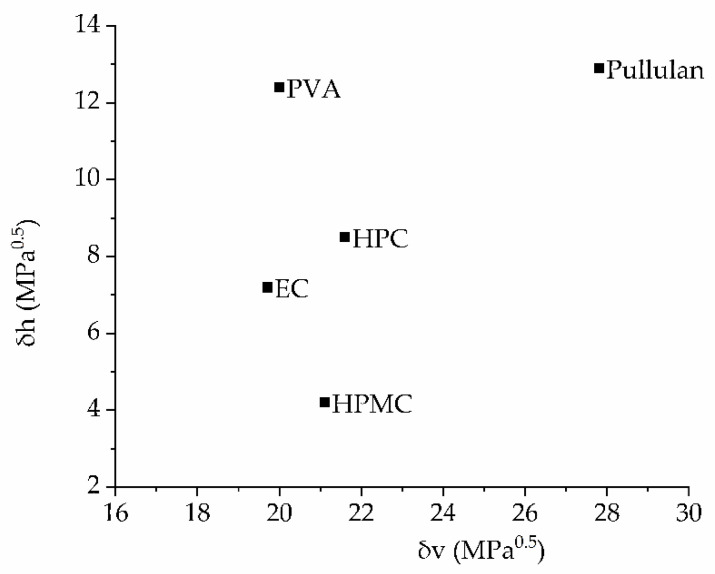
Bagley diagram [32] for the five polymers used for the preparation of tandem films.

**Figure 8 pharmaceutics-14-00264-f008:**
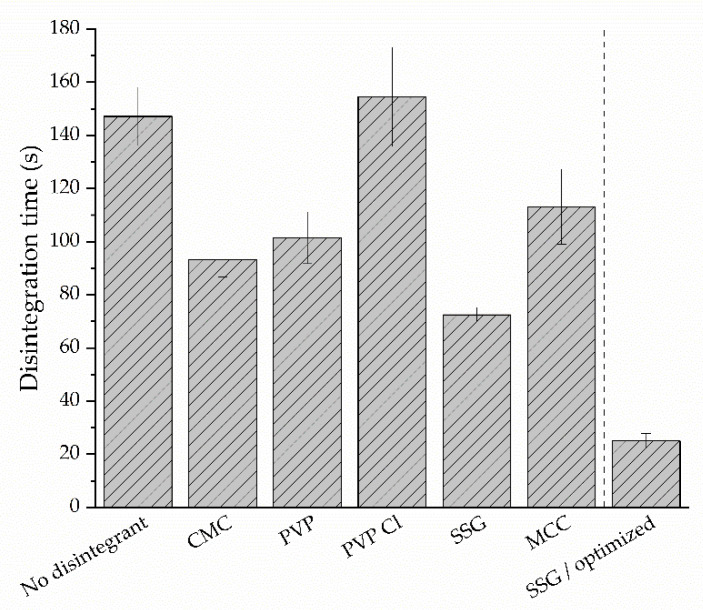
Disintegration times of the pre-formulations and the optimized formulation.

**Figure 9 pharmaceutics-14-00264-f009:**
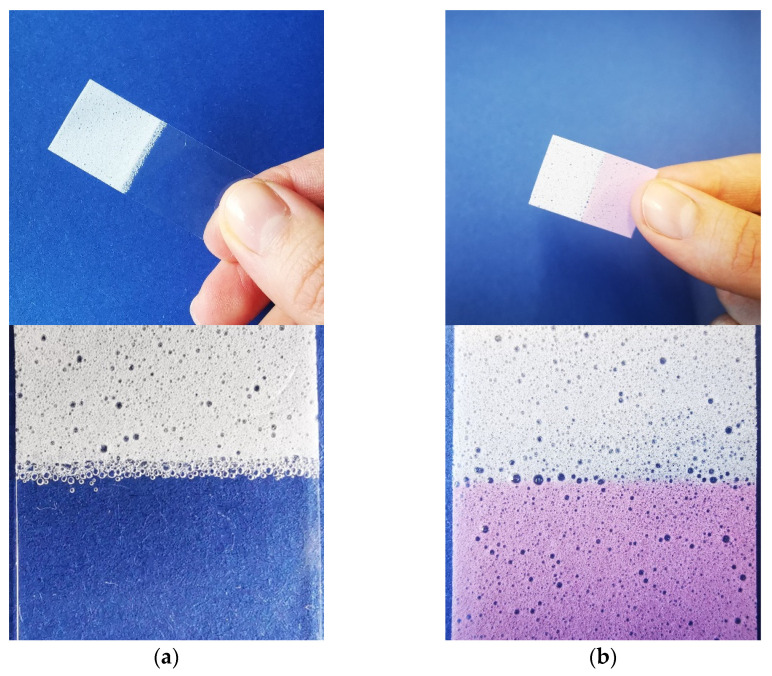
Applicator films with a (**a**) water-insoluble EC applicator and (**b**) water-soluble PVA applicator. The pictures were taken in front of a blue background.

**Figure 10 pharmaceutics-14-00264-f010:**
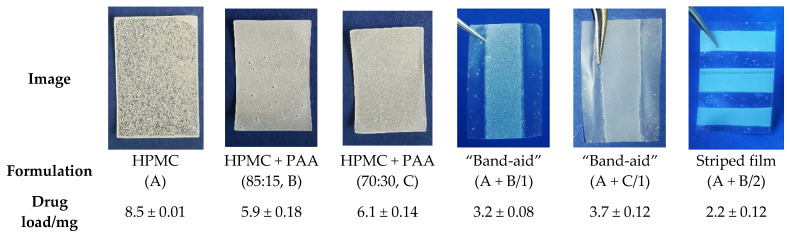
Drug load per film (A: HPMC, B: HPMC + PAA 85:15, C: HPMC + PAA 70:30, *n* = 3–5, mean ± sd). The pictures were taken in front of blue background.

**Figure 11 pharmaceutics-14-00264-f011:**
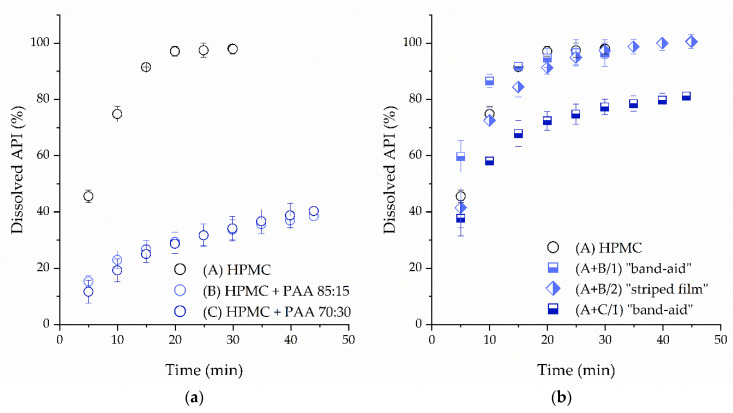
Dissolution profiles of (**a**) mucoadhesive conventional and (**b**) tandem films in comparison to the HPMC “reference” film (*n* = 3, mean ± sd).

**Table 1 pharmaceutics-14-00264-t001:** Placebo solutions for the evaluation of the subdivision models (% *w*/*w*, HV: higher viscosity, LV: lower viscosity, P.606: Pharmacoat^®^ 606, P.603: Pharmacoat^®^ 603).

Substance	HV (Red)	LV (Red)	HV
HPMC (P.606)	15.0	/	15.0
HPMC (P.603)	/	15.0	/
Glycerol	3.0	3.0	3.0
Amaranth solution	0.15	0.15	/
Deion. water	Ad 100.0	Ad 100.0	Ad 100.0

**Table 2 pharmaceutics-14-00264-t002:** Prepared film formulations (selection, % *w*/*w*, N22: Aqualon^TM^ N22).

Formulation Name	Film Former	Plasticizer	Thickening Agent	Solvent
HPMC (P.606)	HPMC (20.0)	Glycerol (4.0)	/	Water
HPC	HPC (20.0)	Glycerol (0.1)	/	Water
PVA	PVA (20.0)	Glycerol (3.0)	/	Water
Pullulan	Pullulan (20.0)	Glycerol (5.0)	Tragacanth (0.2)	Water
EC (N22)	EC (20.0)	Triethyl citrate (5.0)	/	Water (5.0) + ethanol

**Table 3 pharmaceutics-14-00264-t003:** Composition of bisoprolol pre-formulation for disintegration test (% *w*/*w*).

Ingredient	1	2	3	4	5	6	7
HPMC (P.606)	15.0	10.0	10.0	10.0	10.0	10.0	10.0
Disintegrant	/	CMC 5.0	PVP 5.0	PVP Cl 5.0	SSG 5.0	MCC 5.0	SSG 5.0
Glycerol anh.	6.0	6.0	6.0	6.0	6.0	6.0	6.0
Bisoprolol	2.50	2.50	2.50	2.50	2.50	2.50	5.00
Water	Ad 100.0	Ad 100.0	Ad 100.0	Ad 100.0	Ad 100.0	Ad 100.0	Ad 100.0

**Table 4 pharmaceutics-14-00264-t004:** HPLC gradient for quantification of bisoprolol content.

Time (min)	Eluent A (% *v*/*v*)	Eluent B (% *v*/*v*)
3.00	95.0	5.0
11.00	80.0	20.0
19.00	20.0	80.0
20.00	20.0	80.0
22.00	5.0	95.0

**Table 5 pharmaceutics-14-00264-t005:** Polymer solutions for the development of mucoadhesive tandem films (quantities in mass percent, v: verum, p: placebo). Tandem films were cast from combinations of A(v) + B(p) and A(v) + C(p).

Substance	A(v)	B(v)	C(v)	B(p)	C(p)
HPMC (P.606)	10.0	8.5	7.0	8.5	7.0
PAA	/	1.5	3.0	1.5	3.0
Glycerol	4.0	4.0	4.0	4.0	4.0
Theophylline monohydrate	2.0	2.0	2.0	/	/
NaOH	/	q.s. ad pH 5.5	q.s. ad pH 5.5	q.s. ad pH 5.5	q.s. ad pH 5.5
Water	Ad 100.0	Ad 100.0	Ad 100.0	Ad 100.0	Ad 100.0

**Table 6 pharmaceutics-14-00264-t006:** Evaluation of the materials for coating blade inserts based on field experience and literature data (++: excellent, +: good, -: poor, --: absent).

Parameter	Stainless Steel Constructions		3D-Printed Inserts	
	Precursor	Optimized	PLA	ABS
Robustness to thermal and mechanical stress [47,48]	++	++	-	-
Resistance to organic solvents/acids/bases [46,56]	++	++	+	-
Expense of manufacturing	-	-	++	++
Adjustability of compartment sizes	--	--	++	++
Quality of separation	-	+	++	++

**Table 7 pharmaceutics-14-00264-t007:** Properties of the used polymers and respective pre-casting solutions and films prepared for the mechanical characterization of tandem films.

Formulation	Viscosity of the Solution (Pa·s) (Mean ± sd)	Disintegration Time (s)	Dry Film Thickness (µm) (Mean ± sd)	Contact Angle (°) (Mean ± sd)	*δd* (MPa^0.5^)	*δp* (MPa^0.5^)	*δh* (MPa^0.5^)	*δt* (MPa^0.5^)	*δv* (MPa^0.5^)
HPMC	4.2 ± 0.19	44 ± 3.0	70 ± 1.7	63 ± 8.1	16.1	13.5	4.2	21.5	21.1
HPC	1.3 ± 0.08	47 ± 2.4	65 ± 3.0	51 ± 5.1	15.1	15.5	8.5	23.3	21.6
PVA	0.8 ± 0.10	40 ± 5.5	156 ± 4.6	53 ± 2.1	16.8	10.9	12.4	23.5	20.0
Pullulan	2.2 ± 0.17	9 ± 1.2	143 ± 4.0	46 ± 1.5	20.0	19.3	12.9	30.6	27.8
EC	2.5 ± 0.31	Insoluble	73 ± 10.2	64 ± 1.9	16.1	11.4	7.2	21.0	19.7

## Data Availability

The data presented in this study are available in “Fundamental investigations into metoprolol tartrate deposition on orodispersible films by inkjet printing for individualised drug dosing”.

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
