# Peer review of "Concept of Orodispersible or Mucoadhesive “Tandem Films” and Their Pharmaceutical Realization"

_pharmaceutics, 2022, doi:10.3390/pharmaceutics14020264_

Round 1
Reviewer 1 Report
The manuscript provides an overview of the concept in which ways coupled drug-loaded and drug-free thin film segments may be used to provide different benefits in drug administration. The concept applies to both orodispersible and mucoadhesive buccal films. It is an interesting and innovative approach and would be of interest for other researchers working in the field.
The manuscript reports the compilation of screening studies performed, which is somewhat difficult to follow. The authors themselves state that “Different concepts for an application of such tandem films are presented and two of the possibilities are further examined in this publication.” It would be beneficial to clearly define study objectives and hypotheses tested. It would be also beneficial to restructure the manuscript around the two concepts tested, since different approaches, film forming agents and drug substances have been employed.
Conclusions are rather general, and should be re-written to reflect the main findings in line with the re-defined study objectives.
Data presented in Table 4. are not necessary and may be omitted.
Text editing is necessary in order to assure consistency with regards to the way in which polymer names are presented (in tables, particularly), film dimensions units used (cm vs mm), occasional spelling and linguistic errors.
Author Response
Dear reviewer, thank you for your efforts to improve our manuscript. Please see our repsonses in the attachment. Best regards Anja Goebel and Jörg Breitkreutz
Response to Reviewer #1:
The authors are grateful for your time and helpful recommendations. All comments have been taken into account and are answered in detail in the following. For easier reading and understanding, all comments are cited in this document (written in italics) and the comments were subdivided into the single topics that were answered separately. In the revised manuscript, all changes were highlighted by the Track change function.
General assessment
The manuscript provides an overview of the concept in which ways coupled drug-loaded and drug-free thin film segments may be used to provide different benefits in drug administration. The concept applies to both orodispersible and mucoadhesive buccal films. It is an interesting and innovative approach and would be of interest for other researchers working in the field.
Thank you for the overall positive assessment and your suggestions.
Structure
The manuscript reports the compilation of screening studies performed, which is somewhat difficult to follow. The authors themselves state that “Different concepts for an application of such tandem films are presented and two of the possibilities are further examined in this publication.” It would be beneficial to clearly define study objectives and hypotheses tested. It would be also beneficial to restructure the manuscript around the two concepts tested, since different approaches, film forming agents and drug substances have been employed.
When writing the manuscript, these two options for the overall structure (the chosen structure, or a complete separation into the two concepts tested) were weighed against each other. It is preferred to keep the “classical” structure that is also given in the template. We did not want to split the manuscript into two papers and we think that the preferred type of structure is subjective as other reviewers evaluated the research design and structuring as positive. The introduction and conclusion were adapted to make the study objectives and conclusions more clear (paragraph 1 and 4).
Conclusion part
Conclusions are rather general, and should be re-written to reflect the main findings in line with the re-defined study objectives.
The conclusion part has been amended to make the main findings more clear and to point out the technique´s limits and future challenges (paragraph 4).
Table 4 (HPLC gradient)
Data presented in Table 4. are not necessary and may be omitted.
As the HPLC gradient developed differs from the method given in the Ph.Eur., the table was retained for better understanding of the methods used (paragraph 2.10).
Consistency of wording and units; language
Text editing is necessary in order to assure consistency with regards to the way in which polymer names are presented (in tables, particularly), film dimensions units used (cm vs mm), occasional spelling and linguistic errors.
Thank you for your comment – the consistency of language has been re-checked and improved. For example, the word “carbomer” has been completely replaced by “polyacrylic acid” or “PAA” to avoid confusion and the film dimensions (length and width) are described completely in cm now. In Table 1, “Pharmacoat 606” has been changed into “HPMC (P. 606)” as in the other tables and the mucoadhesive tandem film shapes were consistently named as “striped film” or “band aid” shape.

Reviewer 2 Report
The manuscript is well written with appropriate research design with relevant literature. Some clarification can be made are as:
- Did you observe any difference in quality of film if dried at 80 °C
- What could be the reason for the reduction in tensile strength of tandem film formulations compared to the original film formulation
- Any association or correlational study between viscosities of polymer and drug loading and drug release
- Any more modification in this system that can hold more amount of drug and can be tried for non-potent drug
In this manuscript concepts for the preparation of tandem films and their possible applications were developed and their feasibility has been successfully characterized and proven with empirical data.
Author Response
Dear reviewer, thank you for your efforts to improve our manuscript. Please see our repsonses in the attachment. Best regards Anja Goebel and Jörg Breitkreutz
Response to Reviewer #2:
The authors are grateful for your time and helpful recommendations. All comments have been taken into account and are answered in detail in the following. For easier reading and understanding, all comments are cited in this document (written in italics) and the comments were subdivided into the single topics that were answered separately. In the revised manuscript, all changes were highlighted by the Track change function.
General assessment
The manuscript is well written with appropriate research design with relevant literature.
Thank you very much for your positive evaluation and your following suggestions.
Drying temperature
Some clarification can be made are as:
- Did you observe any difference in quality of film if dried at 80 °C
Different drying temperatures were examined during preliminary studies. However, precipitation of HPMC and HPC was observed at higher temperatures due to the cloud point of cellulose ethers, which is about 50 °C for the used HPMC (Pharmacoat 606) and 45 °C for HPC. Hence, a drying temperature of 40 °C was considered to be a good compromise to reach an appropriate drying time and still avoid precipitation. This consideration was added in the results and discussion part (paragraph 3.1).
Tensile strength
- What could be the reason for the reduction in tensile strength of tandem film formulations compared to the original film formulation
The discussion part has been amended (paragraph 3.2).
Viscosity
- Any association or correlational study between viscosities of polymer and drug loading and drug release
Literature research has been conducted but no clear relationship between the viscosities of the polymers / pre-casting masses and the drug release kinetics from the films were found, although a slower dissolution due to higher viscosity would be plausible. On the other hand, increasing the viscosity of polymer dispersions with suspended API obviously increases their stability due to slower sedimentation. Unfortunately, the actual viscosities of the polymer dispersions with suspended APIs could not be measured due to concerns about possible scratches on the rheometer geometries, but increased viscosity was clearly notable during the solvent casting process.
Drug load
- Any more modification in this system that can hold more amount of drug and can be tried for non-potent drug
Although some literature sources describe drug loadings up to 50 mg, I believe that the relatively low drug load is a general limitation of orodispersible and buccal films.
General assessment
In this manuscript concepts for the preparation of tandem films and their possible applications were developed and their feasibility has been successfully characterized and proven with empirical data.
Thank you very much for the positive feedback.

Reviewer 3 Report
In the manuscript »Concept of orodispersible or mucoadhesive “Tandem films” and their pharmaceutical realization« by Anja Göbel and Jörg Breitkreutz, the authors present a modification of the solvent casting technique with the aim to divide oral films into two or more compartments.
The manuscript addresses a lot of issues but sounds well.
Comments:
P6, l 203, Solubility parameter calculations, the calculation of these parameters (dispersion parameter (δd), polarity parameter (δp) 204 and parameter associated with hydrogen bonds (δh)) could be better explained; two given eqs and 2 references seem not to be sufficient. Additionally, in this context, Fig 7 could be better explained.
Language – some minor mistakes could be found, e.g. were instead of was (P 10, L 352).
Author Response
Dear reviewer, thank you for your efforts to improve our manuscript. Please see our repsonses in the attachment. Best regards Anja Goebel and Jörg Breitkreutz
Response to Reviewer #3:
The authors are grateful for your time and helpful recommendations. All comments have been taken into account and are answered in detail in the following. For easier reading and understanding, all comments are cited in this document (written in italics) and the comments were subdivided into the single topics that were answered separately. In the revised manuscript, all changes were highlighted by the Track change function.
General assessment
In the manuscript »Concept of orodispersible or mucoadhesive “Tandem films” and their pharmaceutical realization« by Anja Göbel and Jörg Breitkreutz, the authors present a modification of the solvent casting technique with the aim to divide oral films into two or more compartments.
The manuscript addresses a lot of issues but sounds well.
Thank you very much for the positive assessment of the manuscript.
Solubility parameters
P6, l 203, Solubility parameter calculations, the calculation of these parameters (dispersion parameter (δd), polarity parameter (δp) 204 and parameter associated with hydrogen bonds (δh)) could be better explained; two given eqs and 2 references seem not to be sufficient. Additionally, in this context, Fig 7 could be better explained.
An additional explanation for the calculation of the solubility parameters has been added. Also, the discussion of Figure 7 (Bagley diagram) has been amended (paragraphs 2.5 and 3.2). Reference is made to Just, S.; Sievert, F.; Thommes, M.; Breitkreutz, J., Improved group contribution parameter set for the application of solubility parameters to melt extrusion. Eur. J. Pharm. Biopharm. 2013, 85 (3, Part B), 1191-1199 where the determination of the solubility parameters is presented in detail.
Language
Language – some minor mistakes could be found, e.g. were instead of was (P 10, L 352).
The sentence was split into two sentences to make it more clear (paragraph 3.2).

Reviewer 4 Report
This is a well-organized and well-illustrated research paper, has an important clinical message, and should be of great interest to the readers. The paper reports a modification of solvent casting technique dividing the oral films in to multiple compartments which the aim to improve safety , drug release kineteics and enhance the overall stability of the oral films. Paragraphing is concise and good, and the article consists of important clinical findings and deserve publication after some revisions listed below.
- I sugges the authors to compare the beneficial properties of currently marketed orodispersible films with the modified films that the authors have reported.
- Line number 40 is not clear, please re-write "API release in to the saliva had been realized"?.
- What are the limitations and advantages of tandem films?
- Lines 91 and 92: Aren't oral films mostly used for fast release? Why is it essential to have multiple forms of release kinetics in one film if films are majorly used for the drugs that need faster release kinetics.
- I suggest the authors to include the advantages, challenges and future perspectives for the approval of the proposed film in the conclusions section.
- Please mention the necessary steps for the clinical approval of the film investigated in the current paper in the conclusion section.
Author Response
Dear reviewer, thank you for your efforts to improve our manuscript. Please see our repsonses in the attachment. Best regards Anja Goebel and Jörg Breitkreutz
Response to Reviewer #4:
The authors are grateful for your time and helpful recommendations. All comments have been taken into account and are answered in detail in the following. For easier reading and understanding, all comments are cited in this document (written in italics) and the comments were subdivided into the single topics that were answered separately. In the revised manuscript, all changes were highlighted by the Track change function.
General assessment
This is a well-organized and well-illustrated research paper, has an important clinical message, and should be of great interest to the readers. The paper reports a modification of solvent casting technique dividing the oral films in to multiple compartments which the aim to improve safety , drug release kineteics and enhance the overall stability of the oral films. Paragraphing is concise and good, and the article consists of important clinical findings and deserve publication after some revisions listed below.
Thank you very much for this positive overall assessment of the manuscript and you following suggestions.
Comparison to marketed ODFs
I sugges the authors to compare the beneficial properties of currently marketed orodispersible films with the modified films that the authors have reported.
The amount of marketed films is still low, but it has been made clear now that the bilayer film with layers buffered to diffenent pH values for better stability and tolerability (mentioned in paragraph 1, page 1) is an already approved film. The drawback of this concept are discussed in the following.
Language
Line number 40 is not clear, please re-write "API release in to the saliva had been realized"?.
Thank you for the hint; the sentence has been reworded to make it better understandable (paragraph 1).
Limits / advantages
What are the limitations and advantages of tandem films?
The conclusion has been amended to make clearer the main advantages of the developed films (implementation of the proposed concepts a) and b)). The main disadvantage has been described in the conclusion: “A limitation of this technique remains the limited maximum drug load due to the reduction in the available surface area, which restricts the application to highly potent APIs” (paragraph 4).
Release kinetics
Lines 91 and 92: Aren't oral films mostly used for fast release? Why is it essential to have multiple forms of release kinetics in one film if films are majorly used for the drugs that need faster release kinetics.
Yes, we agree that in most cases, oral films are designed for fast release. Nevertheless, as is described in the introduction, different approached have been tried to design films with prolonged release in the past. However, this is reported as challenging due to the unfavorable surface/volume ratio and raises questions about the patient-friendliness when imagining having a buccal film in the mouth for several hours. Therefore, the practically implemented application consists of a fast-releasing mucoadhesive film, which is more common. Nevertheless, the possibility of prolonged release kinetics / different forms of release kinetics was mentioned for the sake of completeness.
Clinical approval
I suggest the authors to include the advantages, challenges and future perspectives for the approval of the proposed film in the conclusions section.
Please mention the necessary steps for the clinical approval of the film investigated in the current paper in the conclusion section.
The conclusion part has been amended. Currently, the proposed films have been produced as part of academic research, and clinical approval is not yet planned. However, the relevance of repeatable cutting and, thus, dosage uniformity, have been made clear as a future challenge for scale-up (paragraph 4).

Round 2
Reviewer 1 Report
The authors have responded to all the comments and have introduced necessary amendments in the manuscript.